# Numerical simulation of induced cutting in deep coal

Si-fei Liu[1,2], Shuai-feng Lu[1,2], Zhi-jun Wan[1,2], Hong-wei Zhang[1,2] and Ke-ke Xing[1,2]

[1]Key Laboratory of Deep Coal Resource Mining (CUMT), Ministry of Education of China, and [2]School of Mines, China University of Mining and Technology, Xuzhou 221116, People's Republic of China

S-fL, 0000-0001-8331-5355; Z-jW, 0000-0003-2095-6336

mechanical engineering/energy

rock cutting, confining pressure, central cutting, induced cutting, discrete element method

**Author for correspondence:**
Zhi-jun Wan
e-mail: zhjwan@cumt.edu.cn

Deep coal cutting is a hot research topic at present. In this paper, the cutting technology of three-drum shearer was proposed based on previous studies. Besides, the influence of confining pressure on coal cutting performance was studied by using the discrete element method, and the induction effect of central cutting on coal cutting performance was discussed. Moreover, coal cutting with different boundary conditions was simulated with the aid of PFC$^{2D}$ software. The results show that as the confining pressure increases, the model dominated by tensile failure does not change, but the crack gradually develops from the vertical direction to the free surface of coal. The cutting debris first increases and then decreases; so does the cutting force. Under the effect of central cutting, the crack tends to develop towards the free surface of coal more, and both the peak cutting force and the specific energy consumption increase with the increase of confining pressure. Induced by central cutting, with the increase of confining pressure, the reduction value of peak cutting force increases first and then decreases while the reduction value of cutting specific energy consumption increases.

# 1. Introduction

Coal is the main energy source in China. The coal resource buried within 2000 m is up to $5.9 \times 10^{12}$ t, of which more than 50% is buried deeper than 1000 m. These resources are mainly distributed in eastern China where most of the coal mines have entered the stage of deep mining [1–3]. Deep mining is generally characterized by high pressure and strong disturbance [4,5] which both significantly affect the performance of mining machinery. The cuttability of rock, a comprehensive parameter that reflects the interaction between the tool and the rock, is affected by many factors such as rock properties [6,7], mining machinery performance [8,9] and geological conditions [10,11]. At the same time, since geostress is an important geological factor affecting

**Figure 1.** Schematic diagram of mining. (*a*) Ore-body mining and (*b*) tunnel excavation.

**Figure 2.** Three-drum shearer. (*a*) Main view and (*b*) left view.

rock performance, more and more scholars are focusing on deep mechanized mining [12,13]. Gehring [14] found that cutter consumption in deep tunnelling (800 m) was greater than that under lower overburdens. Employing indentation tests and numerical methods, Innaurato *et al.* [15,16] found that rock cuttability decreased by nearly 30% under a confining stress of approximately 10 MPa. Gnirk [17] carried out a series of indentation tests under confining pressures and found that there was a critical confining pressure where the rock exhibited a macroscopic transition from predominantly ductile to the predominantly brittle failure mode. Li *et al.* [18] also obtained the same conclusion. This phenomenon may result from the higher thrust forces required for cutting deep rock under higher confining stresses. However, field observations indicate that the front abutment pressure existing in front of the mining face may cause rock damage and further improve the cuttability of rock [19–21]. In summary, confining pressure is both beneficial and harmful to rock cutting. If the confining pressure can cause the breakage of rock, it is beneficial to rock cutting; otherwise, it will impede the cuttability of rock. Despite the extensive research on the impact of confining pressure on TBM tools, the influence of confining pressure on the pickaxe cutter of shearer during coal mining is rarely investigated. As shown in figure 1, large pressure exists in front of the working face of the deep well [22–24]. Taking the $T_2$ curve as an example, certain stress which remains on the coal surface above the working face prevents the coal from being further damaged, thus seriously hindering the efficient cutting of coal. Therefore, it is necessary to strengthen the understanding of cutting performance of coal in front of the working face with a depth of over 1000 m.

In order to improve the cutting performance of coal in front of working face with a depth of over 1000 m, this paper proposed an induced-cutting mining process in the middle of the coal seam based on the three-drum shearer from the perspective of relieving geostress. Then, a linear single-pick cutting model was established with the aid of discrete element software (PFC$^{2D}$), so that the coal fragment process and cutting force during the cutting process were obtained. Furthermore, the differences in coal cutting mechanisms under confining pressure, central cutting and free boundary condition were analysed, and the variation of cutting force and cutting energy consumption under three boundary conditions were discussed. Finally, the effect of central mining on rock fragmentation and rock cutting was quantitatively analysed.

# 2. Three-drum shearer and mining process

## 2.1. Three-drum shearer

The three-drum shearer consists of three cutting units, namely, the front cutting unit, the intermediate cutting unit and the rear cutting unit (figure 2). The front and rear cutting units are connected to the

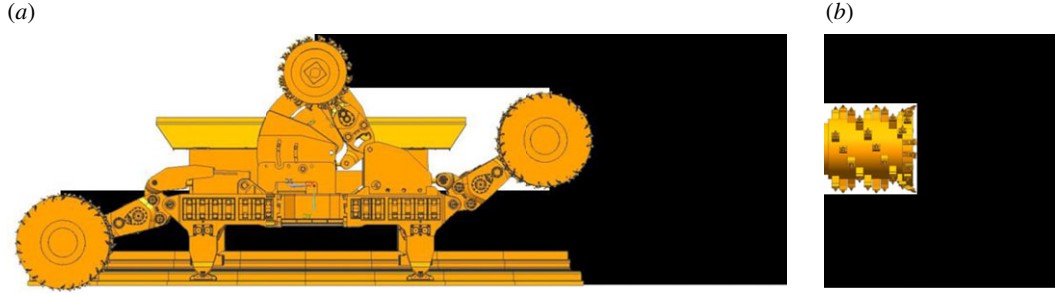

**Figure 3.** Schematic diagram of cutting-induced mining process in the middle. (*a*) Main view and (*b*) left view.

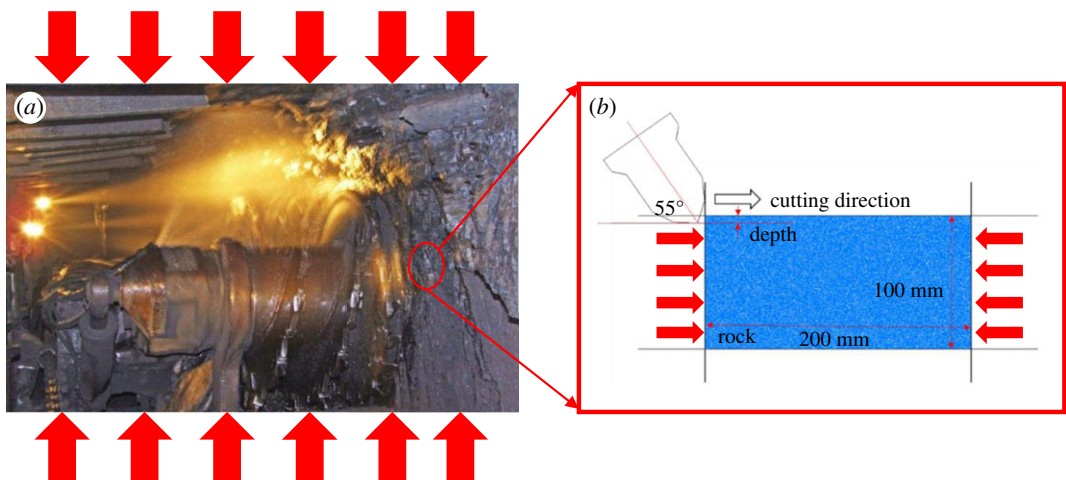

**Figure 4.** Schematic diagram of coal cutting with the drum. (*a*) Confining pressure and (*b*) central cutting models.

traction part at both ends of the fuselage, and heights of the front and rear drums can be adjusted through the upper-mounted height-adjusting cylinders. The intermediate cutting unit is connected to the small frame above the right traction portion, and the intermediate rocker arm can be lifted and lowered through a horizontal height-adjusting cylinder on the shearer.

## 2.2. Central cutting-induced mining process

Based on the principle of slice mining, the mining process of three-drum shearer divides a coal wall into upper, middle and lower layers and then cuts the coal seam by using three drums. First, the front cutting unit cuts the middle coal and opens the coal seam gap. In this way, a free surface is created between the upper coal seam and the lower coal seam, and meanwhile, the pressure of the coal seam is unloaded. Then, the upper and lower coal seams are cut by the intermediate and rear cutting units, respectively, as shown in figure 3.

At the two ends of the working face, the three-drum shearer sweeps the bottom, cleans the coal and realizes oblique cutting through the front and rear drums. At this time, the intermediate cutting unit is adjusted to be flush with the fuselage. Once the front and rear drums have completely entered the coal wall by means of oblique cutting, the intermediate cutting unit starts to be lifted and the intermediate drum gradually cuts into the upper coal seam. At last, the intermediate drum, together with the front and rear drums, enters the regular state of coal cutting to achieve safe recovery of coal seams.

# 3. Coal cutting model

The reasonableness of a numerical model is a guarantee of high-quality results. In order to accurately simulate the induced-mining process of the three-drum shearer, this paper firstly simplified the cutting process of coal wall in the deep working face. As presented in figure 4, the coal-cutting process with the drum was simplified to a single-pick cutting, and the movement of the pick was simplified to a linear one. Next, the cutting boundary conditions were analysed. Considering the

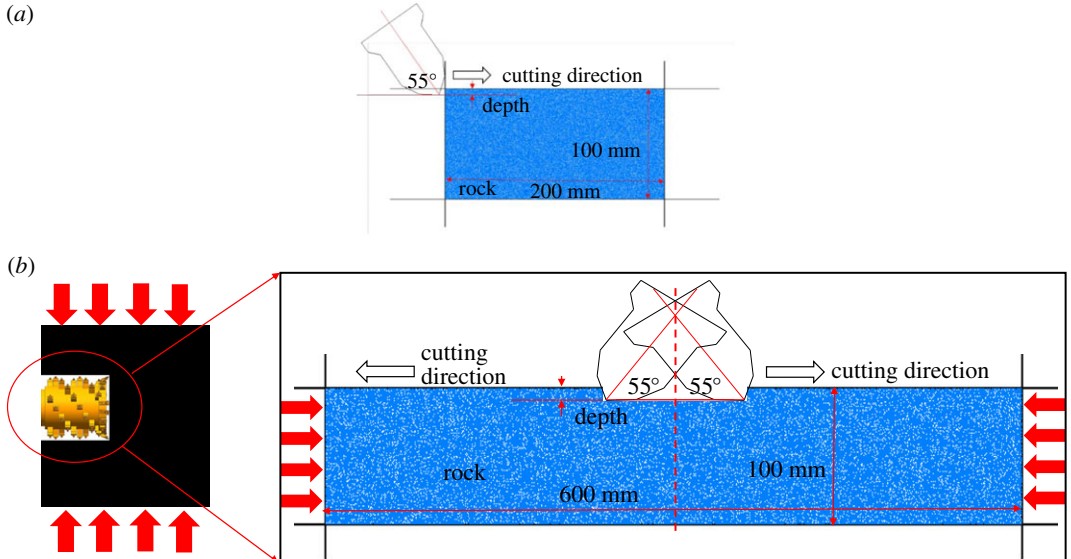

**Figure 5.** Numerical model. (*a*) Confining pressure and (*b*) central cutting models.

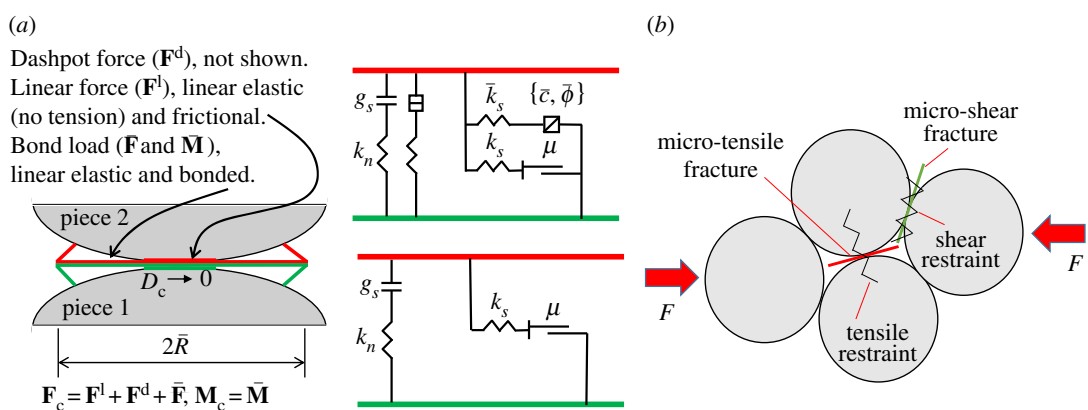

**Figure 6.** Linear parallel bond model. (*a*) Parallel bond model with inactive dashpots and (*b*) schematic diagram of micro-crack formation.

extrusion effect of roof and floor on the coal seam, the cutting of coal wall in the deep working face was simplified to a single-pick cutting model under confining pressure.

Two numerical models, namely, the confining boundary model and the central cutting model, were established, as displayed in figure 5. The first  model which was $200 \times 100$ mm in size was composed of 14 420 particles, while the second one which was $600 \times 100$ mm was composed of 43 440 particles. The coal was cut by a pickaxe cutter with an angle of $55°$, at a speed of $1 \, \mathrm{m \, s^{-1}}$ and for a distance of 120 mm. The cutting depths were 5, 10 and 15 mm. In terms of the boundary condition, the confining boundary model adopted pressure boundaries of 2, 4, 6 and 8 MPa, respectively, and the central cutting model adopted pressure boundaries of 2, 4, 6 and 8 MPa, respectively, and then the central cutting was performed.

The coal sample was taken from a coal seam (with a depth of 1031 m) of Zhaolou Coal Mine of Yanzhou Mining Group. The rock model was established in light of the linear parallel bond model [25]. Parallel bonds, which evenly distribute on the contact interface, can be considered as a series of springs with fixed normal and tangential stiffness between the contact elements. As shown in figure 6*a*, the model mainly involves seven parameters, namely, the elastic modulus of linear contact, the stiffness ratio of linear contact, the elastic modulus of parallel bond, the stiffness ratio of parallel bond, the tensile strength of parallel bond, the shear strength of parallel bond, and the friction angle, which can all be determined by the uniaxial compression test [26,27]. When the rock is damaged, the model will divide micro-cracks into tensile cracks and shear cracks according to whether the maximum stress of a bond

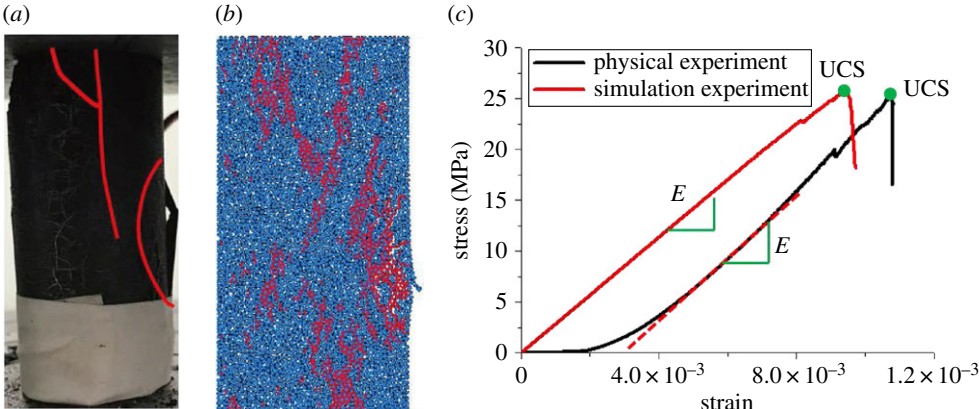

**Figure 7.** Comparison between the physical experiment and the simulation experiment. (*a*) Physical experiment, (*b*) simulation experiment and (*c*) stress–strain curve. UCS, uniaxial compressive strength.

**Table 1.** Model parameters of the linear parallel bond model.

| micro parameters | value | micro parameters | value |
|---|---|---|---|
| particle radius | 0.5–0.75 mm | effective modulus | 1.6 GPa |
| normal stiffness | 1.6 GPa | normal-to-shear stiffness ratio | 2.5 |
| normal-to-shear stiffness ratio | 2.5 | tensile strength (stress) | 8.6 MPa |
| friction coefficient | 45 | cohesion (stress) | 15.2 MPa |

fracturing exceeds the tensile strength or the shear strength of the bond, as can be seen in figure 6*b*. Specific parameters of the coal sample are given in table 1. The results of fitting between numerical simulation and physical experiments are illustrated in figure 7. The results of the Brazilian splitting experiment are shown in figure 8. The Mohr–Coulomb failure line is exhibited in figure 9.

# 4. Results and discussion

## 4.1. Influence of confining pressure on coal cutting

The cutting fragmentation patterns under different confining pressures are exhibited in figure 10, where the cutting debris is displayed in different colours and the micro-cracks are displayed in the form of line segments (red for the tensile crack and green for the shear crack). It can be seen from figure 10 that the size of cutting debris increases first and then decreases with the increase of confining pressure, which is mainly due to the influence of crack expansion. As the confining pressure increases, the expansion of crack in the vertical direction is limited, and it develops toward the free surface, appearing as a shear crack macroscopically. Therefore, when the confining pressure is low, the crack develops in the vertical direction, producing small cutting debris. As the confining pressure increases, the crack gradually expands in the horizontal direction, forming larger debris. When the confining pressure rises again, the crack develops toward the free surface, forming smaller debris again. It can be seen from figure 10 that almost all the micro-cracks belong to tensile cracks.

Figure 11 shows the peak cutting forces (PCF) of coal under different confining pressures. It can be seen from figure 11 that the peak cutting force increases first and then decreases with the increase of confining pressure. This is mainly because the peak cutting force is related to the cutting debris, that is, the formation of cutting debris marks the appearance of peak cutting force [12]. According to the analysis of the law of crack expansion with the change of confining pressure, when there is no confining pressure, the crack develops in the vertical direction, and large cutting debris can hardly be formed, so the peak cutting force is low. When the confining pressure increases gradually (2–6 MPa), the crack develops in the horizontal direction, and it tends to connect to the free surface to generate large cutting debris, so the peak cutting force keeps increasing. When the confining pressure increases

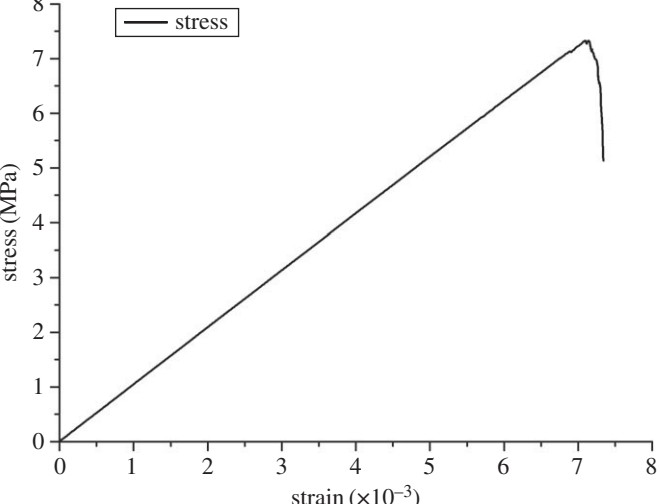

**Figure 8.** Brazilian splitting experiment.

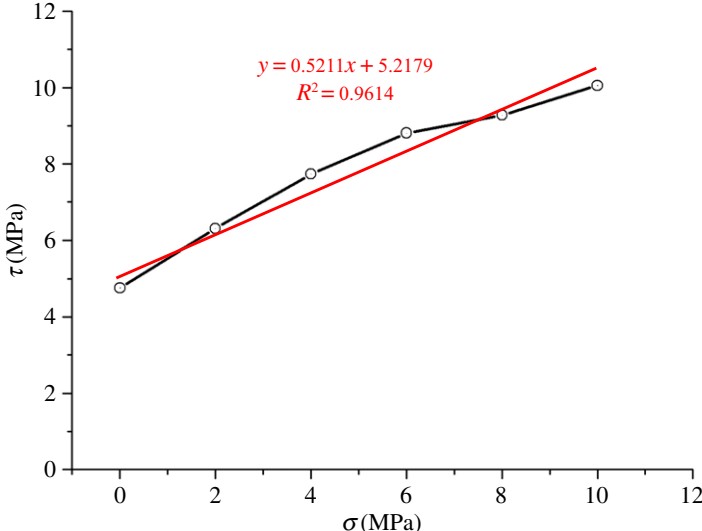

$y = 0.5211x + 5.2179$
$R^2 = 0.9614$

**Figure 9.** Mohr–Coulomb failure line.

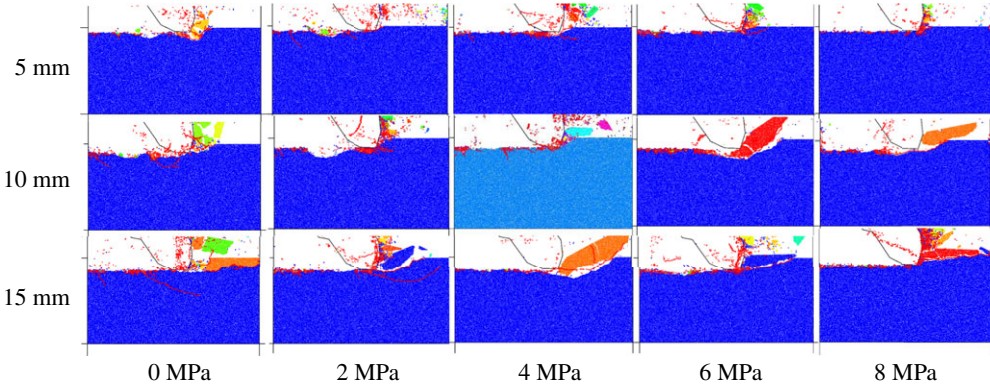

**Figure 10.** Fragmentation patterns of coal under different confining pressures.

to a certain extent (6–8 MPa), the crack develops toward the free surface, and small cutting debris is generated continuously, because the confining pressure further restricts the development of vertical cracks. Therefore, the peak cutting force shows a decreasing trend. In terms of cutting energy

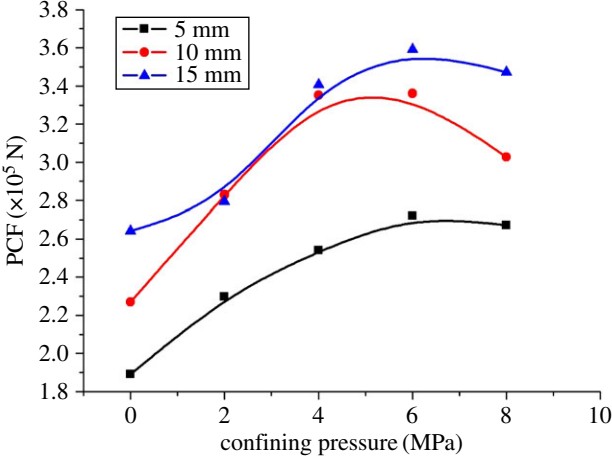

**Figure 11.** Peak cutting force of coal under different confining pressures.

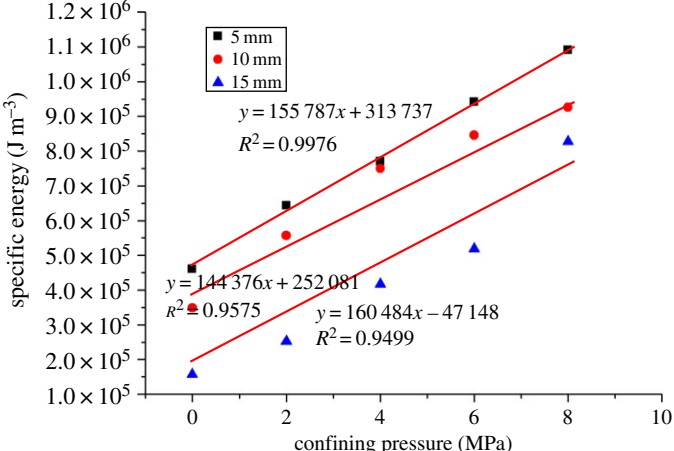

**Figure 12.** Cutting specific energy consumption of coal under different confining pressures.

consumption, the coal–rock cutting specific energy consumption rises linearly with the confining pressure, as shown in figure 12. This is because the strength of coal grows with the increase of confining pressure. According to the theory of energy for rock failure, it takes more energy to break it. Therefore, deep coal cutting requires a greater cutting force and greater cutting energy consumption.

## 4.2. Influence of central cutting on coal cutting

The central cutting model is shown in figure 5$b$. Given the symmetry of the model, this paper only takes half of it as the research object. The results of coal fragmentation are shown in figure 13. As the confining pressure increases, the size of coal debris is significantly reduced. At the same time, the crack gradually develops from the horizontal direction to the free surface, without any crack expansion in the vertical direction. The micro-cracks mainly belong to tensile cracks, with shear cracks accounting for 10% of the total cracks. These results are different from those obtained by other cut models. As presented in figure 14, the peak cutting force of the central cutting model is significantly affected by the confining pressure and increases linearly with it. As can be seen in figure 15, the cutting specific energy consumption of the central cutting model increases linearly with the confining pressure, which is consistent with expectations.

## 4.3. Efficient cutting method of deep coal

As mentioned above, different from the shallow coal, deep coal contains large stress within it that leads to significant changes in cutting fragment, cutting force and cutting energy consumption. Confining

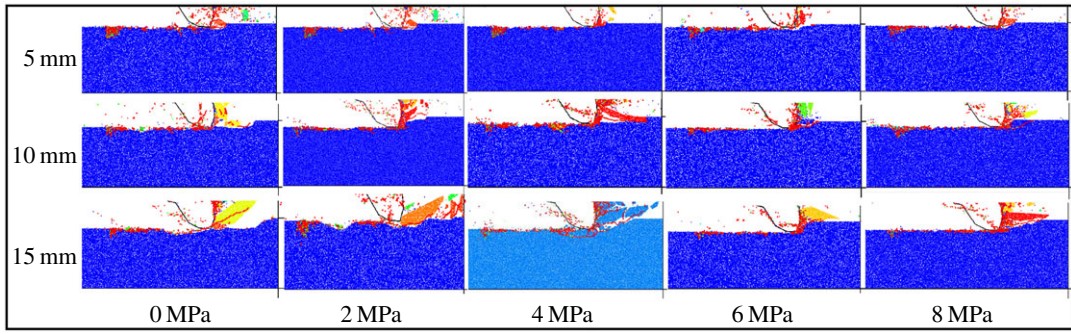

**Figure 13.** Fragmentation patterns of coal under different confining pressures based on central cutting.

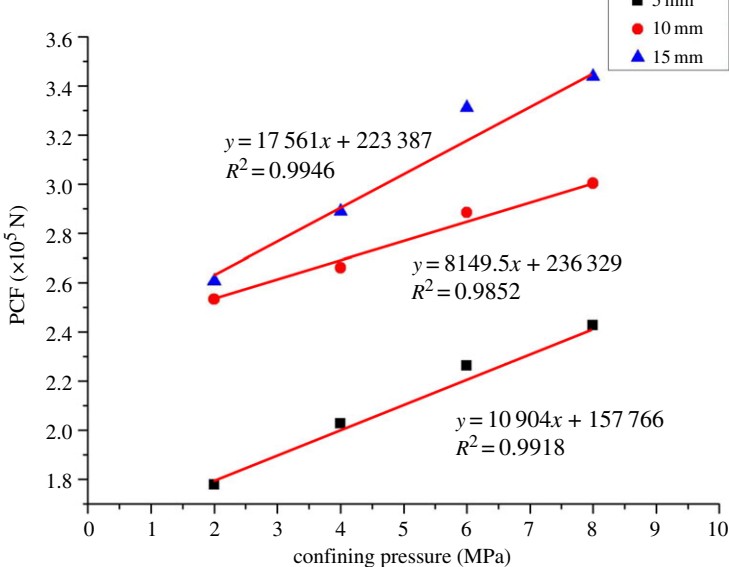

**Figure 14.** Peak cutting force of coal under different confining pressures based on central cutting.

pressure, which can enhance both the strength and the contact force of coal, alters the direction of the minimum principal stress in the coal and limits the development of cracks in the vertical direction. As a result, under the action of confining pressure, the cracks expand in the horizontal direction or to the free surface, thus forming larger cutting debris and increasing the cutting force. In order to unload the internal stress of coal, this paper proposed a central cutting-induced mining process that is able to further strengthen the expansion of cracks to the free surface and reduce the size of cutting debris, as shown in figure 16.

Figure 17 displays the reduction value and ratio of cutting force induced by the central cutting model. The reduction value of peak cutting force increases first and then decreases, because the cutting force is related to both rock strength and crack expansion. When the confining pressure is low, although the confining pressure raises the strength of coal, the central cutting reduces the strength of coal by unloading its internal stress, resulting in a large decrease in the cutting force. When the confining pressure is large, the confining pressure enables the crack to expand to the free surface, while the influence of small-scale central cutting on coal strength is limited, so that the decrease in the cutting force is slight.

Figure 18 is the reduction value and ratio of specific energy consumption induced by the central cutting model. The reduction value increases with the increase of confining pressure. From the perspective of reduction ratio, the reduction ratio is almost constant at 0.29 and 0.35 when the cutting depth is 5 and 10 mm, respectively, whereas it increases when the cutting depth is 15 mm. Therefore, in deep coal seam mining, central cutting-induced mining is a good way to promote cutting efficiency, because it not only reduces the cutting force but also achieves cutting with low energy consumption. Meanwhile, according to the analysis of the experimental results, it is suggested that in field engineering, a medium cutting depth can be adopted when confining pressure is small, as it can

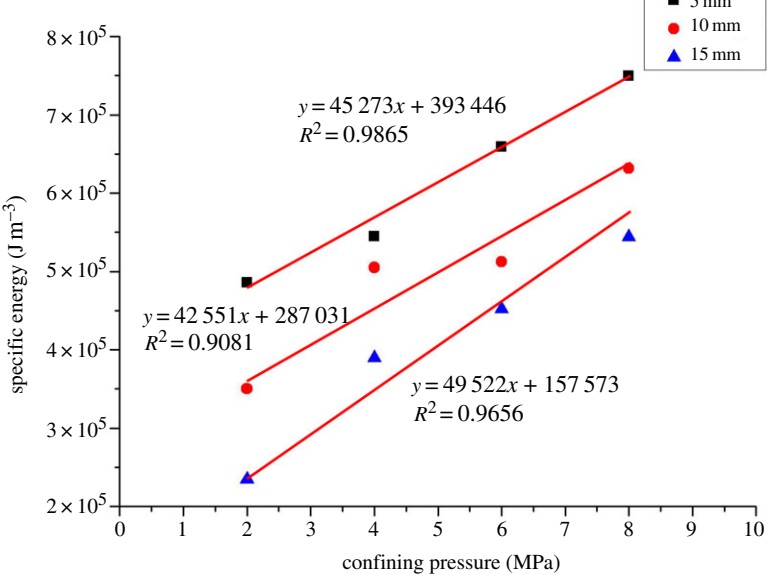

**Figure 15.** Cutting specific energy consumption of coal under different confining pressures based on central cutting.

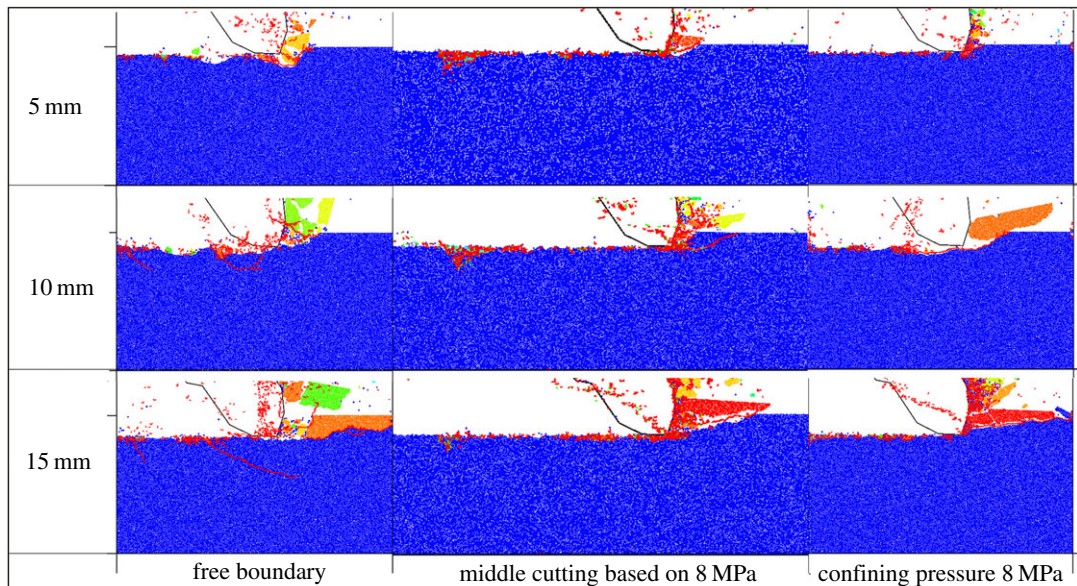

**Figure 16.** Fragmentation patterns of three cutting models.

effectively reduce the cutting force, whereas the cutting depth should be increased when the confining pressure is large, as it can not only reduce the energy consumption but also allow the efficient recovery of coal seams.

## 5. Conclusion

The rock breaking mechanism and cutting efficiency of a single pick are the basis of shearer drum design. In this study, the linear single-pick coal-cutting processes under different boundary conditions were simulated with the aid of discrete element software PFC$^{2D}$. Besides, the effects of confining pressure and central cutting on coal rock fragmentation and cutting performance were analysed. The research results are of great significance to the understanding of rock breaking mechanism of drum and the selection of mining methods for high confining pressure ore bodies. The main conclusions are drawn as follows:

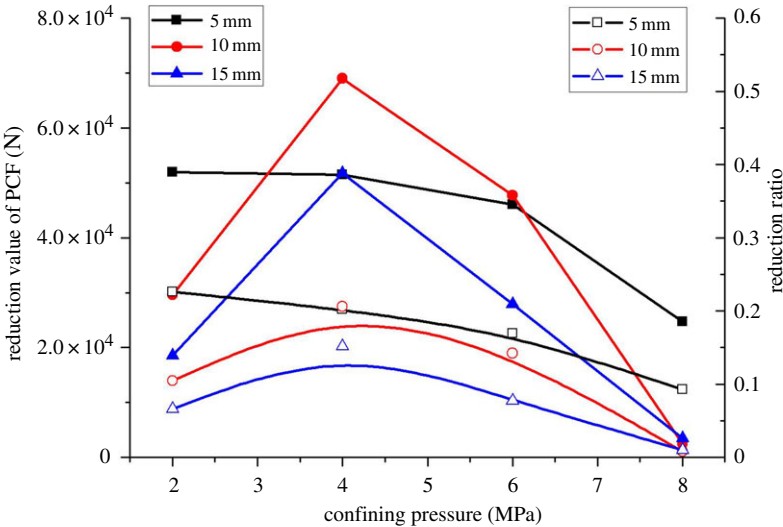

**Figure 17.** Reduction value and reduction ratio of PCF based on central cutting induction.

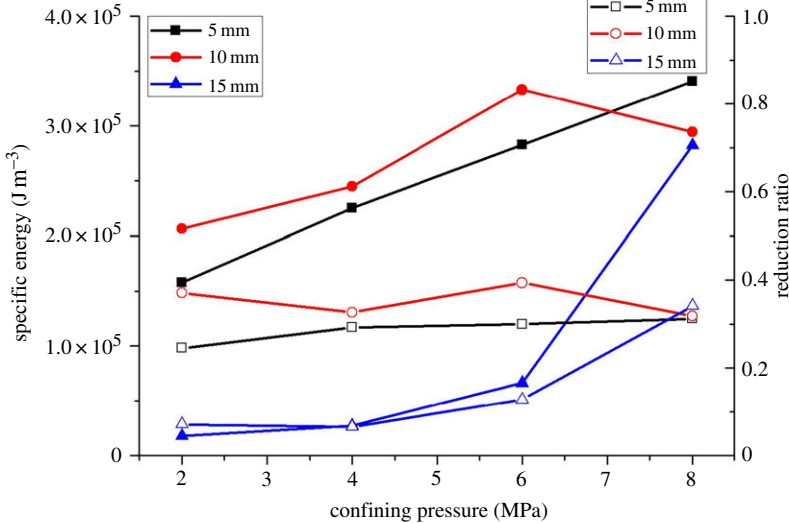

**Figure 18.** Reduction value and reduction ratio of specific energy based on central cutting induction.

(1) A new three-drum shearer and cutting process was proposed. The three-drum shearer consists of three cutting units, namely, the front cutting unit, the intermediate cutting unit and the rear cutting unit. First, the front cutting unit serves to cut the middle section of the coal seam to open a coal seam gap, which creates a free surface for the upper and lower sections of the seam and unloads the coal seam pressure. Then, the upper and lower cutting units are used to cut the upper and lower sections of the seam, respectively.

(2) In terms of rock fragmentation, confining pressure does not change the dominant position of tensile failure in the coal-cutting process, but alters the crack propagation direction. Under the effect of confining pressure, the crack gradually expands from the vertical direction to the horizontal direction and the free surface, and the cutting debris increases first and then decreases. The process of central cutting-induced coal mining is dominated by tensile failure, with only 10% of shear cracks and no crack expansion in the vertical direction. The central cutting method strengthens the tendency of crack development towards the free surface and makes cutting debris smaller.

(3) In terms of cutting force, the peak cutting force, which is related to coal fragmentation, increases first and then decreases with the confining pressure. In the central cutting model, the peak cutting force increases as the confining pressure increases. Under the effect of central cutting, the reduction value of peak cutting force first increases and then decreases with the increase of confining pressure, reaching the maximum reduction value under the confining pressure of 4 MPa. The induction effect of central cutting almost disappears when the confining pressure reaches 8 MPa. At the

same time, the induction effect of central cutting is the most significant at the cutting depth of 10 mm. Therefore, in field engineering, a medium cutting depth can be adopted when the confining pressure is small.

(4) In terms of energy consumption, the cutting specific energy consumption increases with the increase of confining pressure, which is consistent with the fact that the breaking of rocks with larger strength requires greater energy. Under the effect of central cutting, the reduction ratio of cutting specific energy consumption is almost constant at 0.29 and 0.35 at the cutting depth of 5 mm and 10 mm, respectively, whereas it increases when the cutting depth is 15 mm. Therefore, in field engineering, the cutting depth should be increased when confining pressure is high, which can not only reduce energy consumption but also achieve efficient mining of coal seams. The research results of this paper are of great significance for the selection of deep mining methods.

Data accessibility. All original data are deposited at Figshare: https://doi.org/10.6084/m9.figshare.8868572 [28].

Authors' contributions. S.-f.Liu, S.-f.Lu and Z.-j.W. carried out the molecular laboratory work, participated in data analysis, carried out sequence alignments, participated in the design of the study and drafted the manuscript; S.-f.Liu and S.-f.Lu carried out the statistical analyses; S.-f.Liu collected field data; Z.-j.W., H.-w.Z. and K.-k.X. conceived of the study, designed the study, coordinated the study and helped draft the manuscript. All authors gave final approval for publication.

Competing interests. We declare we have no competing interests.

Funding. This research was supported by The Fundamental Research Funds for the Central Universities (grant no. 2017CXNL01).

Acknowledgements. We thank Jingchao Wang and Zhuangzhuang Yang for their support of our study.

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
