## [Reviewer comments · Royal Society Open Science]

Review History

RSOS-190308.R0 (Original submission)

Review form: Reviewer 1

Is the manuscript scientifically sound in its present form?

Yes

Are the interpretations and conclusions justified by the results?

Yes

Is the language acceptable?

Yes

Is it clear how to access all supporting data?

Not Applicable

Do you have any ethical concerns with this paper?

No

Have you any concerns about statistical analyses in this paper?

No

Recommendation?

Reject

Comments to the Author(s)

This paper is similar to the reviewed manuscript before (Investigation for the influence mechanism of rock damage on rock fragmentation and cutting performance by discrete element method). I think it is not proper to accept the similar papers in the same journal.

Review form: Reviewer 2

Is the manuscript scientifically sound in its present form?

Yes

Are the interpretations and conclusions justified by the results?

Yes

Is the language acceptable?

No

Is it clear how to access all supporting data?

Yes

Do you have any ethical concerns with this paper?

No

Have you any concerns about statistical analyses in this paper?

No

Recommendation?

Accept with minor revision (please list in comments)

Comments to the Author(s)

1. The authors mention that the pick cutting simulations were done for a certain coal. In Table 1 they list the micromechanical parameters of the coal but they forgot to mention the macromechanical parameters of this type of rock and demonstrate that their micromechanical model behavior resembles the basis mechanical behavior of the coal.
2. They do not say what was the purpose to prepare a model with rigid boundaries.
3. The use of English language needs some improvements

Review form: Reviewer 3

Is the manuscript scientifically sound in its present form?

Yes

Are the interpretations and conclusions justified by the results?

Yes

Is the language acceptable?

Yes

Is it clear how to access all supporting data?

Yes

Do you have any ethical concerns with this paper?

No

Have you any concerns about statistical analyses in this paper?

No

Recommendation?

Accept with minor revision (please list in comments)

Comments to the Author(s)

The paper is interesting, and it proposes the 3-drum shearer for the cutting of coal in longwall mining. The third arm serves the purpose of relieving the confining stress in deep coal mines in order to decrease the cutting energy consumption and reduce the cutting forces. The authors perform two-dimensional simulations with the commercial PFC2D particle element code. My comments are the following:

1. The use of the English language needs some improvement.
2. I cannot realize the use of the model with prescribed displacements on the boundaries. What is its purpose in the context of this numerical study?
3. The authors do not present the stress-strain curves in compression and tension as well as the Mohr-Coulomb failure line of the specific coal rock they model by using the micromechanical model. They only present the micromechanical parameters in Table 1.
4. They do not present the relation between the size of the coal cuttings with the penetration depth and the spacing among the neighboring cutting picks for each examined case.

Overall the paper is interesting, however some minor revision is needed according to the comments above.

Decision letter (RSOS-190308.R0)

01-Apr-2019

Dear Professor Wan,

The editors assigned to your paper ("Numerical simulation of induced cutting in deep coal") have now received comments from reviewers. We would like you to revise your paper in accordance with the referee and Associate Editor suggestions which can be found below (not including confidential reports to the Editor). Please note this decision does not guarantee eventual acceptance.

Please submit a copy of your revised paper before 24-Apr-2019. Please note that the revision

deadline will expire at 00.00am on this date. If we do not hear from you within this time then it will be assumed that the paper has been withdrawn. In exceptional circumstances, extensions may be possible if agreed with the Editorial Office in advance. We do not allow multiple rounds of revision so we urge you to make every effort to fully address all of the comments at this stage. If deemed necessary by the Editors, your manuscript will be sent back to one or more of the original reviewers for assessment. If the original reviewers are not available, we may invite new reviewers.

- Data accessibility

<http://datadryad.org/submit?journalID=RSOS&manu=RSOS-190308>

- Competing interests

- Authors' contributions

All submissions, other than those with a single author, must include an Authors' Contributions section which individually lists the specific contribution of each author. The list of Authors should meet all of the following criteria; 1) substantial contributions to conception and design, or

acquisition of data, or analysis and interpretation of data; 2) drafting the article or revising it critically for important intellectual content; and 3) final approval of the version to be published.

- Acknowledgements

- Funding statement

on behalf of Professor R. Kerry Rowe (Subject Editor)
openscience@royalsociety.org

Associate Editor's comments:

Thank you for your submission to Royal Society Open Science. We have received 3 reviewer reports. Please carefully address each of the points raised by the reviewers. In particular, you will need to explain the difference between this manuscript and RSOS-190116 (Investigation for the influence mechanism of rock damage on rock fragmentation and cutting performance by discrete element method), as mentioned by reviewer 1.

2 of the reviewers have also suggested that the quality of English needs to be improved. A number of language polishing services are available for authors whose first language is not English. We recommend that you ask a native speaker of English or solicit the support of a language polishing service (<https://royalsociety.org/journals/authors/language-polishing/>) prior to resubmitting the manuscript.

Authors whose papers are returned on language grounds must provide evidence that a professional language editing service or a native speaker of English have assisted in preparing a revised manuscript. Evidence such as a certificate of editing or a signed letter from a native speaker of English would be acceptable.

We look forward to receiving your revised manuscript.

Comments to Author:

Reviewers' Comments to Author:

Reviewer: 1

Comments to the Author(s)

This paper is similar to the reviewed manuscript before (Investigation for the influence mechanism of rock damage on rock fragmentation and cutting performance by discrete element method). I think it is not proper to accept the similar papers in the same journal.

Reviewer: 2

Comments to the Author(s)

1, The authors mention that the pick cutting simulations were done for a certain coal. In Table 1 they list the micromechanical parameters of the coal but they forgot to mention the macromechanical parameters of this type of rock and demonstrate that their micromechanical model behavior resembles the basic mechanical behavior of the coal.

2. They do not say what was the purpose to prepare a model with rigid boundaries.

3. The use of English language needs some improvements

Reviewer: 3

Comments to the Author(s)

The paper is interesting, and it proposes the 3-drum shearer for the cutting of coal in longwall mining. The third arm serves the purpose of relieving the confining stress in deep coal mines in order to decrease the cutting energy consumption and reduce the cutting forces. The authors perform two-dimensional simulations with the commercial PFC2D particle element code. My comments are the following:

1. The use of the English language needs some improvement.
2. I cannot realize the use of the model with prescribed displacements on the boundaries. What is its purpose in the context of this numerical study?
3. The authors do not present the stress-strain curves in compression and tension as well as the Mohr-Coulomb failure line of the specific coal rock they model by using the micromechanical model. They only present the micromechanical parameters in Table 1.
4. They do not present the relation between the size of the coal cuttings with the penetration depth and the spacing among the neighboring cutting picks for each examined case.

Overall the paper is interesting, however some minor revision is needed according to the comments above.

Author's Response to Decision Letter for (RSOS-190308.R0)

See Appendix A.

RSOS-190308.R1 (Revision)

Review form: Reviewer 1

Is the manuscript scientifically sound in its present form?

No

Are the interpretations and conclusions justified by the results?

Yes

Is the language acceptable?

Yes

Is it clear how to access all supporting data?

Yes

Do you have any ethical concerns with this paper?

No

Have you any concerns about statistical analyses in this paper?

Yes

Recommendation?

Major revision is needed (please make suggestions in comments)

Comments to the Author(s)

1. The revised parts cannot be shown in the revised manuscript, I cannot find the revised content.
2. The difference between this paper and the former similar paper has been explained in the proof, however, this explanation is not clear because the internal factors and outer loading process are interacted to influence the mechanical response.

Review form: Reviewer 3

Is the manuscript scientifically sound in its present form?

Yes

Are the interpretations and conclusions justified by the results?

Yes

Is the language acceptable?

Yes

Is it clear how to access all supporting data?

Yes

Do you have any ethical concerns with this paper?

No

Have you any concerns about statistical analyses in this paper?

No

Recommendation?

Accept as is

Comments to the Author(s)

The Reviewers' comments and suggestions have been considered and the revised manuscript can be accepted for publication as is.

Decision letter (RSOS-190308.R1)

04-Jun-2019

Dear Professor WAN:

Manuscript ID RSOS-190308.R1 entitled "Numerical simulation of induced cutting in deep coal" which you submitted to Royal Society Open Science, has been reviewed. The comments of the reviewer(s) are included at the bottom of this letter.

Please submit a copy of your revised paper before 27-Jun-2019. Please note that the revision deadline will expire at 00.00am on this date. If we do not hear from you within this time then it will be assumed that the paper has been withdrawn. In exceptional circumstances, extensions may be possible if agreed with the Editorial Office in advance. We do not allow multiple rounds of revision so we urge you to make every effort to fully address all of the comments at this stage. If deemed necessary by the Editors, your manuscript will be sent back to one or more of the original reviewers for assessment. If the original reviewers are not available we may invite new reviewers.

- Ethics statement

- Data accessibility

- Competing interests

- Authors' contributions

- Acknowledgements

- Funding statement

Kind regards,

Alice Power

Editorial Coordinator

on behalf of R. Kerry Rowe (Subject Editor)

Reviewer comments to Author:

Reviewer: 1

Comments to the Author(s)

1. The revised parts cannot be shown in the revised manuscript,I cannot find the revised content.
2. The difference between this paper and the former similar paper has been explained in the proof, however, this explanation is not clear because the internal factors and outer loading process are interacted to influence the mechanical response.

Reviewer: 3

Comments to the Author(s)

The Reviewers' comments and suggestions have been considered and the revised manuscript can be accepted for publication as is.

Author's Response to Decision Letter for (RSOS-190308.R1)

See Appendix B.

RSOS-190308.R2 (Revision)

Review form: Reviewer 1

Is the manuscript scientifically sound in its present form?

Yes

Are the interpretations and conclusions justified by the results?

Yes

Is the language acceptable?

Yes

Do you have any ethical concerns with this paper?

No

Have you any concerns about statistical analyses in this paper?

No

Recommendation?

Accept with minor revision (please list in comments)

Comments to the Author(s)

The simplified drum cannot represent the real drum, so the cutting process should be deeply investigated and compared with cutting operation of the whole drums. Meanwhile, the research can be applied in the practical drums operation.

Decision letter (RSOS-190308.R2)

07-Aug-2019

Dear Professor Wan,

On behalf of the Editors, I am pleased to inform you that your Manuscript RSOS-190308.R2 entitled "Numerical simulation of induced cutting in deep coal" has been accepted for publication in Royal Society Open Science subject to minor revision in accordance with the referee suggestions. Please find the referees' comments at the end of this email.

The reviewers and Subject Editor have recommended publication, but also suggest some minor revisions to your manuscript. Therefore, I invite you to respond to the comments and revise your manuscript.

- Ethics statement

- Data accessibility

If you wish to submit your supporting data or code to Dryad (<http://datadryad.org/>), or modify your current submission to dryad, please use the following link:
<http://datadryad.org/submit?journalID=RSOS&manu=RSOS-190308.R2>

- Competing interests

- Authors' contributions

- Acknowledgements

- Funding statement

Because the schedule for publication is very tight, it is a condition of publication that you submit the revised version of your manuscript before 16-Aug-2019. Please note that the revision deadline will expire at 00.00am on this date. If you do not think you will be able to meet this date please let me know immediately.

- 1) A text file of the manuscript (tex, txt, rtf, docx or doc), references, tables (including captions) and figure captions. Do not upload a PDF as your "Main Document".
- 2) A separate electronic file of each figure (EPS or print-quality PDF preferred (either format should be produced directly from original creation package), or original software format)
- 3) Included a 100 word media summary of your paper when requested at submission. Please ensure you have entered correct contact details (email, institution and telephone) in your user account
- 4) Included the raw data to support the claims made in your paper. You can either include your data as electronic supplementary material or upload to a repository and include the relevant doi within your manuscript

5) All supplementary materials accompanying an accepted article will be treated as in their final form. Note that the Royal Society will neither edit nor typeset supplementary material and it will be hosted as provided. Please ensure that the supplementary material includes the paper details where possible (authors, article title, journal name).

Kind regards,

on behalf of Professor R. Kerry Rowe (Subject Editor)
openscience@royalsociety.org

Associate Editor Comments to Author:

Thank you for revising the paper further - the reviewer has recommended the paper should be accepted after minor revisions. The Editors would like you to do the following before resubmitting:

- 1) Do a final check for the quality of the written English - you may find the services at <https://royalsociety.org/journals/authors/language-polishing/> to be useful.
- 2) Address the point made in this final reviewer's report that the approach taken cannot currently solve or represent the whole problem - a few final sentences to explore this, and how your work will support future developments would be appreciated.
- 3) Carry out a final check for presentation and clarity.

Reviewer comments to Author:

Reviewer: 1
Comments to the Author(s)

The simplified drum cannot represent the real drum, so the cutting process should be deeply investigated and compared with cutting operation of the whole drums. Meanwhile, the research can be applied in the practical drums operation.

Author's Response to Decision Letter for (RSOS-190308.R2)

See Appendix C.

Decision letter (RSOS-190308.R3)

20-Aug-2019

Dear Professor WAN,

I am pleased to inform you that your manuscript entitled "Numerical simulation of induced cutting in deep coal" is now accepted for publication in Royal Society Open Science.

on behalf of Prof R. Kerry Rowe (Subject Editor)
openscience@royalsociety.org

Appendix A

Revision response to Associate Editor

Response to the comments of associate editor of the paper entitled “ Numerical simulation of induced cutting in deep coal”

Many thanks for constructive comments from the editor and the reviewers. These comments and suggestions have been carefully considered in the current manuscript. Those revised texts were labelled in red in the manuscript. Major changes are summarized as below:

Remark #1: you must include evidence of having had your manuscript reviewed by a language polishing service.

Action: Done. The proof is shown in the figure.

To whom it may concern,

This is to certify that the manuscript entitled “Numerical simulation of induced cutting in deep coal” was edited by the Language Editing Service of Shanghai Wisharing Translation Company. It has adequate English proficiency after the modification.

Detailed information of the manuscript is shown below:

Title: *Numerical simulation of induced cutting in deep coal*

Authors: *Si-fei Liu, Shuai-feng Lu, Zhi-jun Wan, Hong-wei Zhang*

Affiliation: *Key Laboratory of Deep Coal Resource Mining (CUMT), Ministry of Education of China; School of Mines, China University of Mining & Technology, Xuzhou, 221116, China*

We hope you can give a consideration to publication. Please do not hesitate to contact us if any additional information is required.

Yours sincerely,

(General Manager of Shanghai Wisharing Translation Culture Communication Co., Ltd.)

Email: xiaocfanyi@163.com

Tel: +86 13913477610

Remark #2: you will need to explain the difference between this manuscript and RSOS-190116 (Investigation for the influence mechanism of rock damage on rock fragmentation and cutting performance by discrete element method).

Action: Done. The purpose of this study is to investigate the effect of central cutting on coal mining. Firstly, we discussed the inhibition effect of confining pressure on cutting performance. Then, the improved effect of central cutting was analyzed. These inhibition and improved influencers are external factors but not internal. However, the manuscript titled “Investigation for the influence mechanism of rock damage on rock fragmentation and cutting performance by discrete element method” analyzed the internal factors, for example rock properties. In addition, the internal factor aims to improve mechanical design, while the external is focused more on mining method.

Revision response to Reviewer #1

Response to the comments of Reviewer 1 of the paper entitled “

Numerical simulation of induced cutting in deep coal”

Many thanks for constructive comments from the reviewers and the editor. These comments and suggestions have been carefully considered in the current manuscript. Those revised texts were labelled in red in the manuscript. Major changes are summarized as below:

Remark #1: This paper is similar to the reviewed manuscript before (Investigation for the influence mechanism of rock damage on rock fragmentation and cutting performance by discrete element method). I think it is not proper to accept the similar papers in the same journal.

Action: Done. Thanks for comments. The purpose of this study is to investigate the effect of central cutting on coal mining. Firstly, we discussed the inhibition effect of confining pressure on cutting performance. Then, the improved effect of central cutting was analyzed. These inhibition and improved influencers are external factors but not internal. However, the manuscript titled “Investigation for the influence mechanism of rock damage on rock fragmentation and cutting performance by discrete element method” analyzed the internal factors, for example rock properties. In addition, the internal factor aims to improve mechanical design, while the external is focused more on mining method.

Revision response to Reviewer #2

Response to the comments of reviewer 2 of the paper titled “

Numerical simulation of induced cutting in deep coal”

Many thanks for constructive comments from the reviewers and the editor. These comments and suggestions have been carefully considered in the current manuscript. The revised texts were marked in red in the manuscript. Major changes are summarized as below:

Remark #1: The authors mention that the pick cutting simulations were done for a certain coal. In Table 1 they list the micromechanical parameters of the coal but they forgot to mention the macromechanical parameters of this type of rock and demonstrate that their micromechanical model behavior resembles the basis mechanical behavior of the coal.

Action: Done. The results of uniaxial compression test are added, as shown in Figure 7, and details of the PFC model calibration are also shown in the figure.

Remark #2: They do not say what was the purpose to prepare a model with rigid boundaries.

Action: Done. Upon your reminding, we carefully reviewed the research content of this paper and found that rigid boundaries model was not discussed in this paper. At the same time, the research content of rigid boundaries model has nothing to do with the content of this paper, so we decide to delete it.

Remark #3: The use of english language needs some improvements.

Action: Done. Errors have been corrected after polishing the language.

Revision response to Reviewer #3

Response to the comments of reviewer 3 of the paper titled “ Numerical simulation of induced cutting in deep coal”

Many thanks for the constructive comments from the reviewers and the editor. These comments and suggestions have been carefully considered in the current manuscript. Those revised texts were marked in red in the manuscript. Major changes are summarized as below:

Remark #1: The use of the English language needs some improvement..

Action: Done. Thanks for comments. Errors have been corrected after polishing the language.

Remark #2: I cannot realize the use of the model with prescribed displacements on the boundaries. What is its purpose in the context of this numerical study?

Action: Done. Upon your reminding, we carefully reviewed the research content of this paper and found that this model was not discussed in this paper. At the same time, the research content of this model has nothing to do with the content of this paper, so we decide to delete it.

Remark #3: The authors do not present the stress-strain curves in compression and tension as well as the Mohr-Coulomb failure line of the specific coal rock they model by using the micromechanical model. They only present the micromechanical parameters in Table 1.

Action: Done. In this paper, stress-strain curves in compression and tension were added as well as the mohr-coulomb failure line of the specific coal rock, as shown in Figure 7, Figure 8 and Figure 9 respectively.

Remark #4: They do not present the relation between the size of the coal cuttings with the penetration depth and the spacing among the neighboring cutting picks for each examined case.

Action: Done. The spacing among the neighboring cutting picks is an important factor

affecting the cutting effect. A two-dimensional model is used to simulate the cutting process of single pick in this paper, and the two-dimensional model can not simulate the influence of pick spacing on the cutting effect.

Appendix B

Revision response to Reviewer #1

Response to the comments of Reviewer 1 of the paper entitled “ Numerical simulation of induced cutting in deep coal”

Many thanks for constructive comments from the reviewers and the editor. These comments and suggestions have been carefully considered in the current manuscript. Those revised texts were labelled in red in the manuscript. Major changes are summarized as below:

Remark #1: The revised parts cannot be shown in the revised manuscript, I cannot find the revised content.

Action: Done. We are very sorry that you couldn't see the revised part. Now the revised part has been marked in red. According to the reviewers' comments, we have added the references and the basic parameters calibration and polished the language.

Remark #2: The difference between this paper and the former similar paper has been explained in the proof, however, this explanation is not clear because the internal factors and outer loading process are interacted to influence the mechanical response.

Action: Done. The internal factors here refer to the mechanical properties of rocks, while the external conditions are different environments of rocks, as shown in the figure. Environmental factors (such as in-situ stress, confining pressure and temperature) and internal factors (such as elastic modulus, Poisson's ratio and strength) are two concepts in mechanics and it can be studied independently. In our paper, more attention is paid to environmental factors, that is to say, the internal properties (e.g. elastic modulus,

strength, etc.) of rocks have not changed, but the experimental stress conditions have been changed. However, in former similar papers mentioned by the reviewer, the experimental stress conditions have not changed, but the properties of rocks have changed. Besides, they focus on the effect of rock properties on cutting performance. Therefore, our experimental purposes and designs are different.

In addition, the most important thing is that in similar article (according to the reviewer), only the layout design of cutting head is studied, while our paper focuses on the mining method of deep coal seam, and puts forward an induced-cutting mining process in the middle of coal seam based on the three-drum shearer.

To sum up, there is no good correlation between the two articles.

Revision response to Reviewer #3

Response to the comments of Reviewer 1 of the paper entitled “

Numerical simulation of induced cutting in deep coal”

Many thanks for constructive comments from the reviewers and the editor. These comments and suggestions have been carefully considered in the current manuscript. Those revised texts were labelled in red in the manuscript. Major changes are summarized as below:

Remark #1: The reviewers' comments and suggestions have been considered and the revised manuscript can be accepted for publication as is.

Action: Done. Thank you for your comments.

Appendix C

Dear Editors:

We have completed the revision of all comments from reviewers.

Here, I will answer one question.

The purpose of this study is to investigate the effect of central cutting on coal mining. Firstly, we discussed the inhibition effect of confining pressure on cutting performance. Then, the improved effect of central cutting is analyzed. These inhibition and improved influencers are external factors but not internal. However, the manuscript titled “Investigation for the influence mechanism of rock damage on rock fragmentation and cutting performance by discrete element method” analyzed the internal factors, for example rock properties. That is to say, the internal factor aims to improve mechanical design, while the external is focused more on mining method.

If you have any queries, please don't hesitate to contact me at the address below.

Thank you and best regards.

Yours sincerely,

Wan Zhijun

Corresponding author:

Name: Zhijun Wan

E-mail: zhjwan@cumt.edu.cn